# Genome-Wide Identification and Expression Analysis of SOS Response Genes in *Salmonella enterica* Serovar Typhimurium

**DOI:** 10.3390/cells10040943

**Published:** 2021-04-19

**Authors:** Angela Mérida-Floriano, Will P. M. Rowe, Josep Casadesús

**Affiliations:** 1Departamento de Genética, Facultad de Biología, Universidad de Sevilla, Apartado 1095, E-41080 Sevilla, Spain; angmerflo@gmail.com; 2Institute of Microbiology and Infection, University of Birmingham, Birmingham B15 2TT, UK; w.rowe@bham.ac.uk

**Keywords:** SOS response, LexA, LexA box, heterology index, phenotypic heterogeneity

## Abstract

A bioinformatic search for LexA boxes, combined with transcriptomic detection of loci responsive to DNA damage, identified 48 members of the SOS regulon in the genome of *Salmonella enterica* serovar Typhimurium. Single cell analysis using fluorescent fusions revealed that heterogeneous expression is a common trait of SOS response genes, with formation of SOS^OFF^ and SOS^ON^ subpopulations. Phenotypic cell variants formed in the absence of external DNA damage show gene expression patterns that are mainly determined by the position and the heterology index of the LexA box. SOS induction upon DNA damage produces SOS^OFF^ and SOS^ON^ subpopulations that contain live and dead cells. The nature and concentration of the DNA damaging agent and the time of exposure are major factors that influence the population structure upon SOS induction. An analogy can thus be drawn between the SOS response and other bacterial stress responses that produce phenotypic cell variants.

## 1. Introduction

The SOS regulon is a bacterial gene network that facilitates survival to DNA damage [1]. Transcription of SOS genes is under the control of the LexA repressor, which binds the so-called “LexA box” or “SOS box”, a cognate DNA motif whose sequence varies among different species [2,3]. In the absence of DNA damage, LexA prevents transcription of SOS genes [4]. Upon DNA damage, the recombination protein RecA forms nucleofilaments on single-stranded DNA. RecA nucleoprotein filaments activate autoproteolysis of LexA, and transcription of the SOS response genes is turned on [4,5]. Events that produce single-stranded DNA include arrest or stalling of DNA replication, collision between transcription and replication machines, formation of R-loops, aggression by DNA-damaging agents and DNA uptake during transformation or conjugation [5].

Genes belonging to the SOS regulon encode DNA repair functions, DNA polymerases able to perform translesion DNA synthesis, proteins involved in cell division control and additional gene products with ancillary roles in genome defense [4,5,6]. SOS activation also triggers induction of prophages and transcription of genes encoding bacteriocins [7]. The composition of the SOS regulon differs from one bacterial genus to another [5,8,9,10,11].

In *Salmonella enterica* serovar Typhimurium, homologs of *E. coli* SOS genes were described in the 1980s [12,13,14]. Additional members of the SOS regulon were identified with a genetic screen [15] and by PCR-assisted RNA fingerprinting [16]. Comparison of the *Salmonella* and *E. coli* SOS responses reveal strong (but not absolute) conservation [15,17]. Differences have been also described among *S. enterica* strains: for instance, *lexA* null mutations are lethal in the historic strain LT2 due to induction of the Fels2 prophage [18] but are viable in other model strains. A relevant aspect of the *S. enterica* SOS response is the involvement of DNA repair and DNA recombination functions in the interaction with the animal host [19,20,21,22,23]. In fact, inhibition of the SOS response has been considered a potential therapeutic strategy [24].

In this study, a combination of bioinformatic and transcriptomic analysis has identified 48 genes belonging to the *S. enterica* SOS regulon. All genes in the list are canonical in the sense that they harbor LexA boxes and undergo transcriptional activation in response to DNA damage. Their expression patterns, however, turn out to be complex due to the formation of SOS^OFF^ and SOS^ON^ subpopulations, a phenomenon observed both in the absence of DNA damage and upon SOS response activation.

## 2. Materials and Methods

### 2.1. Bacterial Strains, Culture Media and Growth Conditions

All *Salmonella enterica* strains used in this study belong to serovar Typhimurium. Unless indicated otherwise, the strains derive from ATCC 14028 (Appendix A). Strain ST4/74 [25] was used in specific experiments. The source of the *lexA* (Ind^−^) allele was strain DA6522 [18], kindly provided by John R. Roth, University of California, Davis, CA, USA. The *recD* and *recF* alleles were described by Cano et al. [20]. Chromosomal mutations were transduced between strains using phage P22 HT 105/1 *int201* [26]. The P22 transduction protocol and the procedure for isolation of phage-free transductants were described elsewhere [27].

Liquid cultures were prepared in lysogeny broth (LB) and were grown at 37 °C with shaking at 200 rpm. LB solid medium contained agar at a final concentration of 15 g/L. Chemicals were used at the following concentrations: ampicillin, 100 μg/mL; chloramphenicol, 20 μg/mL; kanamycin, 50 μg/mL; nalidixic acid, 8.5 μg/mL; hydroxyurea, 0.1 M; mitomycin C, 0.25 μg/mL; H_2_O_2_, 0.01%. Green plates [28] contained methyl blue (Sigma-Aldrich, St Louis, MO, USA) instead of aniline blue.

### 2.2. RNA Isolation, cDNA Library Preparation and Illumina Sequencing

Overnight cultures were diluted 1:100 in 5 mL of LB. For the ‘non-inducing’ condition the three biological replicates were incubated at 37 °C, 200 rpm until early stationary phase (ESP, OD_600_ ~2.00). Three ‘SOS-inducing’ condition replicates were incubated 1 h in LB, and nalidixic acid was added to a final concentration of 8.5 μg/mL. The cultures were incubated 4 more hours before RNA extraction (ESP, OD_600_ ~1.60).

Total RNA was isolated using the Direct-zol RNA MiniPrep Plus kit (Zymo Research, Irvine, CA, USA). For RNA-seq, cDNA libraries were prepared and sequenced by Vertis Biotechnologie AG (Freising, Germany). The indexed sequencing libraries were pooled in equimolar amounts, size-selected to 200–550 bp and sequenced on an Illumina NextSeq 500 system using 75 bp read length.

### 2.3. Read Processing and Alignment

The quality of RNA-seq libraries was assessed using FastQC v0.11.5 (Babraham Institute, Cambridge, UK) (http://www.bioinformatics.babraham.ac.uk/projects/fastqc (accessed on 20 September 2017)). Trimmomatic v0.36 tool (The Usadel Lab, Aachen, Germany) was used to remove the Illumina TruSeq adapter sequences, leading and trailing bases. Reads with a length shorter than 40 nucleotides after trimming were discarded for further analysis. The remaining reads of each library were aligned to the sequence of the published *S.* Typhimurium ATCC 14028 chromosome (accession: CP001363.1) using Bowtie2 v2.2.29 [29], and alignments were filtered with Samtools v1.3.1 [30] using a MAPQ cut-off of 15. The complete RNA-seq pipeline used for this study is described in https://github.com/will-rowe/rnaseq (accessed on 27 November 2017).

### 2.4. Quantification of Absolute Gene Expression Levels

Absolute gene expression levels were calculated as Transcript Per Million (TPM) values [31,32] and were generated for protein coding genes and noncoding sRNAs in the chromosome of *S.* Typhimurium ATCC 14028. Based on the values obtained, the expression cut-off was set as TPM > 10 for both types of genes [33].

### 2.5. Differential Gene Expression Analysis with Three Biological Replicates

Raw read counts from the three biological replicates in ESP and the three biological replicates in ‘SOS-inducing’ conditions were uploaded into Degust (http://degust.erc.monash.edu/ (accessed on 27 November 2017)). Data were analyzed using the Voom/Limma approach [34,35] with an FDR of ≤ 0.001 and a Log_2_FC ≥ 1. This cut-off was set to identify genes slightly induced by nalidixic acid. To remove genes with low counts, thresholds of ≥ 10 read counts and ≥ 1 Counts Per Million (CPM) in at least the three biological replicates of one sample were used [36].

### 2.6. Identification of LexA Boxes

Putative LexA boxes were identified using the FIMO motif scanning tool [37]. The consensus LexA box used as a motif was 5‘ TAC TGT ATA TAT ATA CAG TA 3′ [38], scanned against the *S.* Typhimurium ATCC 14028 chromosome (accession number CP001363.1) using default parameters. Both DNA strands were scanned using default parameters, and only matches with *p*-value < 0.0001 were selected.

### 2.7. Construction of Transcriptional GFP Fusions

A fragment containing the promoterless green fluorescent protein (*gfp*) gene, originally amplified from pZEP07 [39], and a kanamycin resistance cassette, originally amplified from pKD13 [40], was amplified with primers listed in Appendix A. These primers added flaking regions homologous to the 3′-UTR of the selected genes. The PCR products were integrated into the *Salmonella* chromosome using the Lambda Red recombination system [40]. Since the homologous regions are located downstream the gene stop codon, the integration creates a transcriptional *gfp* fusion and the upstream gene remains intact.

### 2.8. Flow Cytometry

Flow cytometry was used to monitor gene expression at the single cell level. For both inducing and non-inducing conditions, cultures were grown in LB at 37 °C. For SOS-inducing conditions, inducers were added during exponential growth. Flow cytometry assays were performed using bacterial cells diluted in PBS 1x. For discrimination of live and dead cells, propidium iodide was used [41]. Data acquisition was performed using a Cytomics FC500-MPL cytometer (Beckman Coulter, Brea, CA, USA). Data were collected for 100,000 events per sample, and were analyzed with FlowJo 8.7 software (Tree Star Inc. Ashland, OR, USA).

### 2.9. Statistical Analysis

Student’s *t*-test (unpaired) was performed using GraphPad Prism version 6.0 for Mac to determine the statistical differences between two groups.

## 3. Results

### 3.1. Identification of LexA Boxes in the Chromosome of S. enterica Serovar Typhimurium

Search for the consensus LexA binding sequence, 5’ TAC TGT ATA TAT ATA CAG TA 3’ [38] using the software tool FIMO [37] identified 318 potential LexA boxes with a fairly regular distribution in both DNA strands across the chromosome (Appendix A). Unlike other studies that identified putative LexA boxes [2,42], no additional parameters were applied. Despite the high number of putative LexA boxes identified, FIMO analysis failed to detect at least two known LexA boxes: the second box in the *lexA* gene [2] and the third box in the *recN* gene [2]. These boxes were manually added.

### 3.2. Identification of SOS Response Loci

The list of putative LexA-regulated genes was narrowed down by identifying loci that altered their expression under SOS-inducing conditions. For this purpose, we used Illumina-Seq to analyze the transcriptome of *S.* Typhimurium ATCC 14028 in the presence of sub-lethal concentrations of nalidixic acid, a fluoroquinolone that induces the SOS response by targeting DNA gyrase and topoisomerase IV [43,44]. An average of 11,010,991 reads per sample were obtained for non-inducing conditions, and an average of 35,550,086 reads per sample for SOS-inducing conditions. Raw and processed data from RNA-Seq analysis have been deposited at the Gene Expression Omnibus (GEO) database (http://www.ncbi.nlm.nih.gov/geo/ (accessed on 7 October 2020)), with accession number GSE159310. Using a Log_2_FC ≥ 1, FDR = 0.001 cut-off, 1091 upregulated loci and 1073 downregulated loci were detected (Appendix A) Downregulated loci include metabolic, flagellar and chemotaxis genes as well as loci belonging to *Salmonella* pathogenicity islands 1 and 2. Downregulation was not investigated further.

SOS regulon candidates were shortlisted by identifying genes that were upregulated in the presence of nalidixic acid and harbored a SOS box at maximum distance of 400 base pairs upstream of the start codon. This cross-reference identified 47 genes (Table 1), of which 25 had been identified previously as SOS response genes either *S. enterica* or *E. coli* [2,9,42]. The *tisB* and *istR-1,2* genes, which encode the TisB-IstR toxin antitoxin system, had also been described as LexA-regulated [45]. Our analysis also identified prophage genes known to be regulated by LexA, such as the *dinI* homologues *STM14_3210, STM14_1156 and STM14_1439* encoded by Gifsy-1, Gifsy-2 and Gifsy-3, respectively [46]. Other prophage-borne candidates identified in our analysis were *STM14_3214* and *STM14_1432,* which encode phage replication proteins of Gifsy-1 and Gifsy-3, respectively [47]. Genes that were not known to be part of the SOS regulon were also identified as loci under putative LexA control. Examples include *cysP*, which encodes a thiosulfate-binding protein [48]; the phosphate metabolism gene *yqaB* [49]; and *hupA*, which encodes the alpha subunit of the nucleoid-associated protein HU [50]. Certain candidate genes that are divergently transcribed appeared to share an SOS box with a neighbour (e.g., *ssb* and *uvrA*, and *nlhH* and *higB-2*).

Thirty-one genes that harbored a LexA box at a maximum distance of approximately 100 bp were chosen for further analysis. Genes co-induced with adjacent canonical SOS regulon gene(s) through transcriptional read-through [8,51] were not included (e.g., *yebF* and *yebE*). The SOS gene *ruvA* was manually added to the list as it had not appeared among the loci upregulated by nalidixic acid. Genes *ftsY* and *ybfE*, identified as SOS genes by Erill et al. [2], were upregulated by nalidixic acid but their SOS boxes were not identified in our search.

### 3.3. Single Cell Analysis of SOS Gene Expression in the Absence of SOS Induction

To analyze at the single cell level the expression patterns of the SOS genes under study, a transcriptional fusion with the green fluorescent protein (*gfp*) was constructed downstream of the stop codon of each gene. After preliminary validation, certain genes were excluded from further analysis due to lack of fluorescence (*dinG, ydjQ* and *ydjM*) or because flow cytometry failed to detect induction by nalidixic acid (*cysP* and *gudD*). Failure to detect expression does not rule out that these genes may be part of the SOS regulon. In the remaining genes under study, flow cytometry analysis detected heterogeneous expression in LB (that is, in the absence of any known DNA damaging agent). Three main gene groups can be tentatively distinguished:(i).Group I, which is the most numerous (12 loci) presents bistable expression: the bacterial population contains a major subpopulation in which the gene is completely OFF and a smaller subpopulation of cells in which the gene is expressed (ON). The size of the ON subpopulation varies from one gene to another, ranging from ~8.7% in *sulA* to 1.3% in *dinP* and 0.8% and *nlhH*. In all loci, the ON subpopulation is absent in *lexA* (Ind^−^) and *recA* backgrounds. Expression of *lexA*, *recA*, *umuDC* and colicin genes in non-inducing conditions has been described previously in *E. coli* [52,53,54], and the formation of ON subpopulations under such conditions has been proposed to be triggered by activation of the SOS response upon spontaneous DNA strand breakage [53]. In agreement with this view, our analysis of *gfp* fusions did not detect ON cells in a *recA* background (Figure 1), which seems to rule out the possibility that the subpopulation might be produced by spontaneous alleviation of LexA repression (if that were the case, an ON subpopulation should be still detected in a *recA* mutant). Furthermore, the size of the ON subpopulation increased in a *recD* background and to a lesser extent in a *recF* background (Appendix A), in agreement with the major role played by the RecBCD recombination pathway in double-strand break repair [55,56].(ii).Group II genes show “noisy”, heterogeneous expression that splits the population into OFF and ON subpopulations. The percentages of ON cells are larger than in genes of group I (from ~13% in *uvrD* up to ~85% in *ssb*). However, the main difference with group I is that formation of ON cells remains unaltered in *lexA* (Ind^−^) and *recA* backgrounds (Figure 2). We thus conclude that expression of these genes under non-inducing conditions is not under LexA control. Because this group contains housekeeping genes, a tentative interpretation is that the genes may be active in a subpopulation of cells in the absence of DNA damage. It is also possible that cells in the OFF state might have expression levels below the threshold for experimental detection.(iii).Genes classified into group III show heterogeneous expression that is not bimodal, and two ON subpopulations are detected. However, these subpopulations differ in their expression level, and only the ON subpopulation with higher expression level disappears in a *lexA* (Ind^−^) mutant (Figure 3). The latter observation has a paradoxical side as formation of the “high ON” subpopulation appears to be LexA-dependent. Interestingly, the *lexA* gene belongs to this group. Expression of *lexA* in a *lexA* (Ind^−^) background was not tested because a *lexA::gfp* fusion could not be constructed.

Loci that show noisy expression with a pattern opposite to that of group III are also detected (Appendix A). In these loci, a subpopulation with higher expression levels appears in a *lexA* (Ind^−^) background only. No gene in this group has been previously reported as a member of the SOS regulon; however, they harbor a LexA box and are activated by DNA damage. Therefore, one cannot discard that they may have SOS-related functions that remain to be identified [57].

A SOS response gene with a unique expression pattern above is *recA* (Appendix A). Expression in virtually all cells is consistent with the pleiotropic role of RecA in genome maintenance.

### 3.4. Influence of the Distance and the Nucleotide Sequence of the LexA Box on the Expression Patterns of SOS Genes

The existence of various expression patterns under non-inducing conditions led us to investigate the underlying cause. For this purpose, we made the reductionist hypothesis that pattern divergence might be caused by differences in the LexA box. When we examined the distance between the LexA box and the start codon of the cognate gene, we found that bistable expression under LexA control (found in groups 1 and 3) correlates with shorter distances (21.6 ± 5.8 bp). In contrast, genes whose bistability is LexA-independent (group 2) tend to have their LexA box further away (58.8 ± 11.6 bp) (Figure 4A). In genes where the transcription start site (TSS) is known [33], the LexA boxes of genes with LexA-dependent bistability overlap with the TSS. This analysis was made omitting prophage genes, and *ssb* and *uvrA* were also excluded because their LexA box is shared.

The sequence of LexA boxes was also examined as their relatedness to the consensus determines the binding affinity of the LexA repressor [58]. Divergence from the consensus LexA box was measured using a mathematical formula described by Berg and von Hippel [38,59], and the resulting score was expressed as an ‘heterology index’ (HI) as defined by Lewis et al. [60]. Genes that presented LexA-dependent bistability had lower HI values (6.29 ± 0.87), while higher HI’s (10.04 ± 1.1) were common among genes with LexA-independent bistability (Figure 4B). Altogether, these observations indicate that both the location and the sequence of the LexA box determine the expression pattern of the cognate gene under non-inducing conditions.

After observing an influence of the heterology index of the LexA box on the gene expression pattern, we wondered if there was any relationship between the predicted LexA affinity and the size of the subpopulation of cells that expressed the gene under non-inducing conditions. Contrary to our expectations, genes with lower HI (and therefore with predicted higher LexA affinity) were those that presented larger ON subpopulations (Figure 5).

Because this observation was puzzling, we took advantage of the existence of a single nucleotide polymorphism in the LexA box of the *yebG* gene in *S. enterica* ATCC 14028 (the strain used in this work) and another model strain, ST4/74 (Figure 6A). The SNP, a G→C substitution, lowers the HI of the *yebG* SOS box of ST4/74 down to 4.98, compared with 7.03 in ATCC 14028. When we analyzed *yebG*::GFP expression by flow cytometry, the ON subpopulation was higher in ST4/74 ON, thus strengthening the unsuspected observation that genes with LexA-dependent bistability produce larger ON subpopulations in the absence of DNA damage if the LexA box has a low HI (Figure 6B).

To further probe this unsuspected conclusion, the SNPs were swapped in both strains. Single cell analysis revealed that a *yebG*_ATCC 14028_ gene with an SNP_ST4/74_ produced more cells in the ON state, while the introduction of SNP_ATCC 14028_ into the *yebG*_ST4/74_ gene lowered the percentage of ON cells (Figure 6C). Hence, the nucleotide sequence of the SOS box can influence the size of the ON subpopulation under non-inducing conditions in a counterintuitive manner: one would expect a larger ON subpopulation in genes with higher HI values (in other words, in genes whose SOS boxes are predicted to have less affinity for the LexA repressor). The actual observation is, however, the opposite.

We also examined whether a correlation might exist between the noisy pattern of gene expression in groups II and III and the heterology index of their SOS boxes. No correlation was found (Appendix A).

### 3.5. Gene Expression Patterns of S. enterica SOS Genes upon Activation of the SOS Response

To monitor gene expression under SOS inducing conditions, cultures were exposed to DNA damaging agents nalidixic acid [44], hydroxuyrea [61], mitomycin C [62] and oxygen peroxide [63]. Propidium iodide (PI), a DNA-binding dye for which only cells with a damaged membrane are permeable [41], was used to detect dead cells. In the heat maps obtained by flow cytometry analysis, four types of bacterial cells could thus be detected: SOS^OFF^ alive (GFP^−^ PI^−^), SOS^OFF^ dead (GFP^−^ PI^+^), SOS^ON^ alive (GFP^+^ PI^−^) and SOS^ON^ dead (GFP^+^ PI^+^). The loci chosen for these experiments belonged to groups I, II and III as well as to the miscellaneous group showing individual gene expression patterns, and representative examples are shown in Figure 7. Relevant observations can be summarized as follows:(i).The locus-specific expression patterns detected in non-inducing conditions largely disappeared upon SOS induction, with minor differences that did not correlate with the group classification (Figure 7). Differences were, however, seen depending on the DNA inducer, which seems to be a major determinant of the expression pattern.(ii).A large subpopulation of live cells (GFP^+^ PI^−^) with an active SOS response was detected in all loci under study.(iii).Subpopulations made of GFP^+^PI^+^ cells were also detected. The fact that such cells had an active SOS response but were a PI^+^ suggests that they may have a compromised cell membrane. Hence, they may be tentatively considered dead or bound to die.(iv).Subpopulations of dead cells that did not show SOS induction (GFP^−^ PI^+^) were also detected. A subpopulation of this kind was especially conspicuous upon treatment with nalidixic acid. A tentative interpretation is either that SOS induction did not take place in such cells or that SOS induction failed to tolerate DNA damage.(v).Detection of subpopulations that did not show GFP nor PI fluorescence admits more than one explanation. One is that repair of DNA damage is highly efficient in such cells, thus permitting that the SOS response is rapidly turned off. This explanation seems, however, unlikely, as a relatively stable GFP variant has been used [39]. An alternative possibility is that the non-fluorescent subpopulation is made of cells that survive DNA damage without inducing the SOS response, and an attractive speculation is that they might be in a dormant state. However, we cannot ignore the limitations of using PI as indicator of cell death: as a DNA binding agent, if the cell has lost its genetic material due to severe damage it will not be stained and will appear in the “live” fraction (GFP^−^ PI^−^).(vi).When flow cytometry data were collected to represent cell sizes in the *y*-axis (instead of PI fluorescence as in Figure 7), filamentation was detected at various extents, especially when hydroxyurea or nalidixic acid were used as inducers (Appendix A).

Additional single cell analysis was performed to monitor the expression pattern of SOS genes upon different times of exposure to a given inducer. The example shown in Figure 8 involves *yebG*, a gene of group I. Different patterns of *yebG* expression were detected in the presence of low and high concentrations of nalidixic acid. An enigmatic bistable pattern with formation two YebG^ON^ subpopulations was detected at a high concentration, and the sizes of the subpopulations changed over time. The significance of latter observation is unknown.

The *yebG* gene was also used as a model to investigate the evolution in the number of live/dead cells over time. As shown in Figure 9, dead cells were more abundant in the YebG^OFF^ subpopulation and their number increased over time. However, only live cells were detected at 24 h and the population was made of both YebG^OFF^ and YebG^ON^ cells. Absence of dead cells is an enigmatic observation, especially in the YebG^OFF^ subpopulation. A hypothetical explanation, speculative at this stage, is that activation of defense mechanisms other than the SOS response may protect YebG^OFF^ cells from nalidixic acid. Selection of mutants is one possibility. Efflux pump activation [64] and other non-mutational mechanisms are also conceivable.

## 4. Discussion

Our search for *S. enterica* genes that are upregulated by DNA damage and harbor a LexA box at or near the promoter region identified 48 loci, 25 of which had previously been described as SOS response genes in either *Salmonella* or *E. coli* [2,9,42] (Table 1 and Appendix A). The list of loci shared with *E. coli* includes *recA, recN, sulA, polB, umuDC*, *smbC*, *istR*, *tisB*, various *din* and *uvr* genes, and the *lexA* gene itself. Novel *S. enterica* SOS loci identified in this study include prophage homologues of *dinI*, genes whose role in genome defense is unknown (*corE, cysP, deoD, frsA, gudD, higB, hupA, ndh, ndlH, rcnB, slyX* and *yqaB*) and loci of unknown function (*yacA*, *yejK* and *yhjE*). Involvement in DNA repair may seem odd for certain genes of the list; however, possession of LexA box(es) and activation by DNA damage may argue in favor of an unknown role in DNA repair. Indeed, we cannot exclude the possibility that some of the “odd” gene products may play a unknown role in the SOS response besides their known activity [57].

To monitor the expression of SOS genes, transcriptional *gfp* fusions were constructed directly on the chromosome, downstream of the stop codon of each gene. This method generates fusions that leave the main gene body intact. Another advantage is that the fusions are constructed at the native genomic location of the gene under study, thus avoiding instability, copy number variation and other potential problems of expression analysis using plasmids. Furthermore, use of flow cytometry to monitor gene expression permits the examination of a large number of cells (typically 100,000), a population size significant enough to detect subpopulations even if they are small.

Detection of both SOS^OFF^ and SOS^ON^ cells in the absence of external DNA damage confirms previous observations made by fluorescence microscopy in the *lexA*, *recA*, *umuDC* and colicin genes of *E. coli* [52,53,54]. Our observations show that heterogeneous expression is a feature of many, perhaps all SOS genes. The existence of SOS^ON^ subpopulations in LB cultures suggests that endogenous DNA damage may occur in certain cells, thus causing SOS activation. Absence of the SOS^ON^ subpopulation in a *recA* background supports this interpretation, which is also in agreement with the literature: endogenous formation of reactive oxygen species is known to be common during normal metabolism [65,66], and spontaneous DNA strand breakage is detected during normal growth [53].

Genome sequence analysis reveals that genes with LexA-dependent bistability tend to have their SOS boxes closer to the coding sequence, often overlapping with the transcription start site (Table 1). Another trait of genes with LexA-dependent bistability is that their SOS boxes show low heterology indexes (Figure 4), which suggests higher affinity towards LexA [60]. On the contrary, the SOS boxes of genes with LexA-independent bistability are farther from the start codon and have higher HI’s (Figure 4). These traits are indicative of weaker interaction with LexA [60], thus providing a plausible explanation for LexA independence.

In contrast with the above observations, which can be discussed in a canonical scenario, a paradox of LexA-dependent bistability is that genes whose SOS boxes have lower HI’s present larger ON subpopulations in the absence of DNA damage (Figure 5). Swapping of the *yebG* SNP in strains ATCC 14028 and ST4/74 also swaps the ON subpopulation sizes (Figure 6), thus confirming an inverse correlation between HI and ON subpopulation size. We leave the paradox unsolved. A tentative speculation, however, is that the SOS boxes of certain genes may have evolved to combine high LexA affinity with the ability to allow swift derepression. The relatively large sizes of the ON subpopulations under non-inducing conditions may support this view.

Genes of groups II and III are activated by DNA damage but have a basal level of expression which is independent of SOS induction. The existence of LexA-independent heterogeneity seems to indicate the involvement of additional factors, which is not surprising if one considers that some such genes encode housekeeping proteins (e.g., the DNA replication protein Ssb [67]). The list also includes recombination (*ruvA*) and repair genes (*uvrA*, *uvrB* and *uvrD*) that are involved in mechanisms of maintenance of DNA integrity independent of SOS [68,69]. Hence, a tentative explanation for the differences in LexA dependence under non-inducing conditions is that genes that play non-SOS roles present basal expression not controlled by LexA. In contrast, genes that are exclusively involved in the SOS response are tightly repressed by LexA, and expression in a fraction of cells reflects spontaneous SOS activation. In support of this view, *sulA, umuDC, dinP (dinB)* and prophage genes present the latter expression pattern.

Subpopulations were also detected upon SOS activation, and dead cells were common. However, the expression patterns of live cells were less complex and more amenable to interpretation than under non-inducing conditions. A major factor that shapes the population structure is the nature of the DNA damaging agent (Figure 7), an observation consistent with the notion that each chemical may cause DNA damage in a specific manner. Under the conditions employed in this study, nalidixic acid and mitomycin C elicited stronger SOS responses. In turn, nalidixic acid and hydroxyurea caused more cell death than MMC and H_2_O_2_ (Figure 7). Additional factors that appear to modulate the population structure are the concentration of inducer (Figure 8) and the time of exposure (Figure 9). Timing of induction, a feature not investigated here, has been also shown to be under the influence of the LexA binding kinetics at SOS boxes [70].

Formation of certain types of cell variants cannot be considered a genuine property of the SOS response, and a relevant example is cell death. The fact that not all the cells die certainly underscores the individuality of bacterial cells but does not involve any SOS-associated attribute [71]. In other cases, however, cell-to-cell differences appear to arise from specific features of the promoters and/or the LexA boxes of SOS genes. For instance, a feature of loci classified in groups II and III is that formation of OFF and ON cells does not result from a bimodal gene expression pattern but from heterogeneous, “noisy” gene expression that produces ON cells above a threshold [72]. Mathematical modeling suggests that gene expression noise is not a mere consequence of stochastic interactions between the transcriptional machinery and promoters but a trait that has evolved to optimize the workings of regulatory networks [73]. In the case of the SOS response, noise may facilitate gradual activation in the response to damage, thus introducing a property of analogue circuits into the digital LexA-based control. This combination of analogue and digital elements may increase robustness [74,75].

The descriptive nature of this study makes restraint advisable in the interpretation of the observations made. However, an incontestable conclusion may be that phenotypic heterogeneity adds an additional layer of complexity to the SOS response. This view is in agreement with previous studies that detected expression of *E. coli* SOS genes in non-inducing conditions [52,53,54]. Furthermore, cell-to-cell differences in DNA repair responses other than the SOS response have been described. For instance, heterogeneity of spontaneous DNA replication errors has been shown to generate subpopulations of *E. coli* cells with increased mutation frequencies [76]. Another example involves antibiotic-induced activation of the RpoS-dependent general stress response, which increases the mutation rate in a subpopulation of cells [77]. Variation of mutation rates has been also observed in *E. coli* cells that undergo stochastic activation of the adaptive response to DNA alkylation damage [78]. Cell-to-cell heterogeneity may thus be a common feature of DNA repair systems [79]. Because of the relevance of DNA repair for survival and evolution, formation of bacterial cell variants may increase the adaptive capacity of the population under environmental threat [80,81,82,83]. If this view is correct, an analogy can be drawn between the SOS response and other bacterial stress responses that produce phenotypic cell variants [75,84].

## Figures and Tables

**Figure 1 cells-10-00943-f001:**
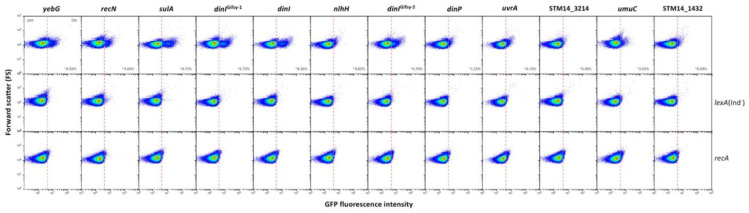
LexA-dependent bistability of group I loci in the absence of known DNA damage. The dashed line shows the boundary between OFF and ON states, established as the expression level of a non-fluorescent sample (Appendix A).

**Figure 2 cells-10-00943-f002:**
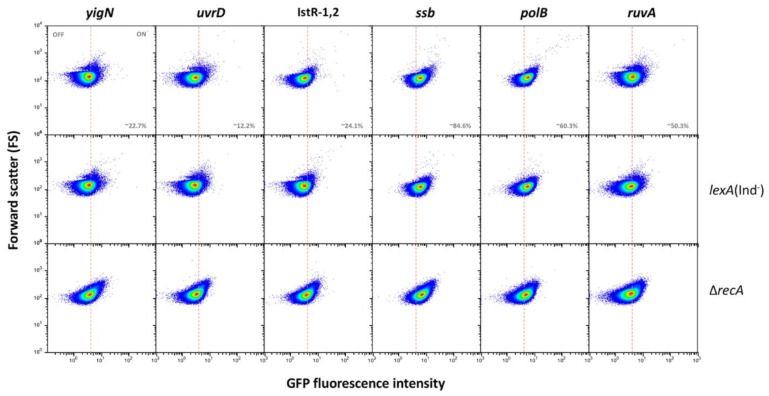
LexA-independent gene expression heterogeneity in group II genes. The dashed line shows the boundary between OFF and ON states, established as the expression level of a non-fluorescent sample (Appendix A).

**Figure 3 cells-10-00943-f003:**
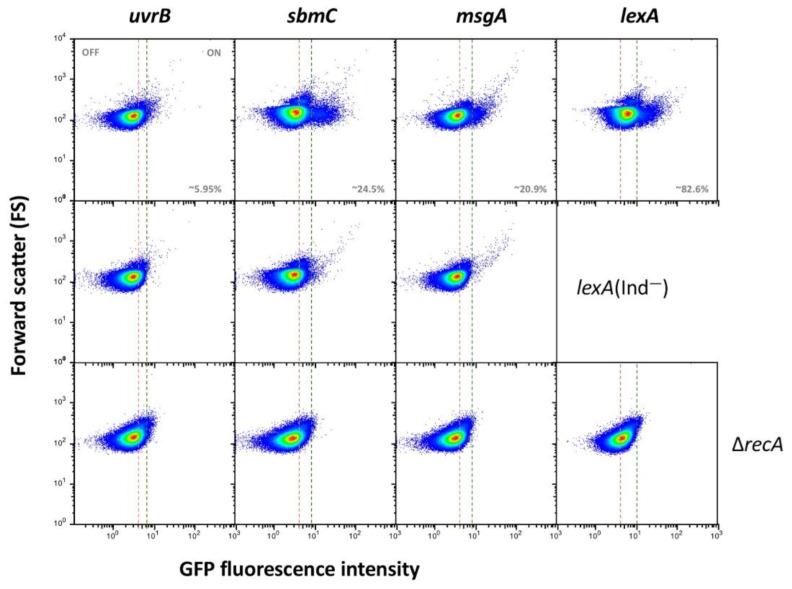
Formation of LexA dependent and LexA-independent ON subpopulations in genes of group III. The pink dashed line shows the boundary between OFF and ON states, established as the expression level of a non-fluorescent sample (Appendix A).

**Figure 4 cells-10-00943-f004:**
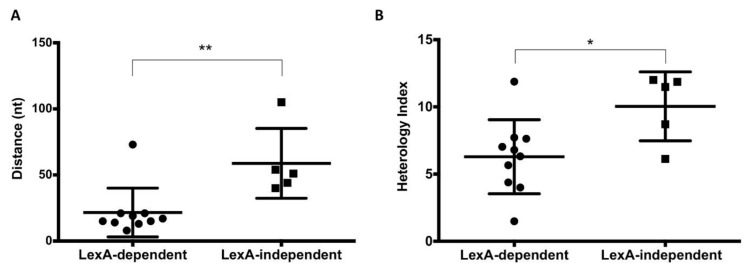
(**A**). Correlation between LexA-dependent and LexA-independent bistability and the distance of the SOS box. Statistically significant differences are shown (*p*-value: * *p* < 0.05; ** *p* < 0.01). (**B**). Correlation between LexA-dependent and LexA-independent bistability and the heterology index (HI) of the SOS box. In genes that show LexA-dependent bistability and have more than one LexA box (*recN*, *dinI* and *lexA*), only the closest LexA box with lower HI was considered.

**Figure 5 cells-10-00943-f005:**
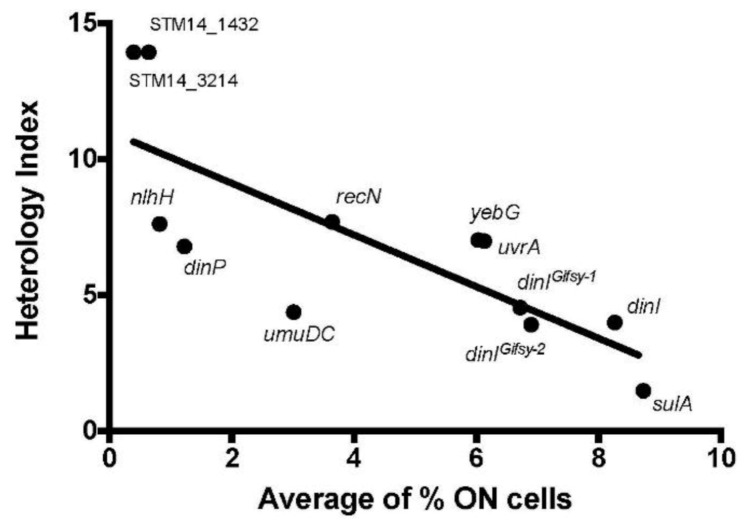
Correlation between the HI values of the SOS boxes and the size of the ON subpopulations of genes with LexA-dependent bistability (group I).

**Figure 6 cells-10-00943-f006:**
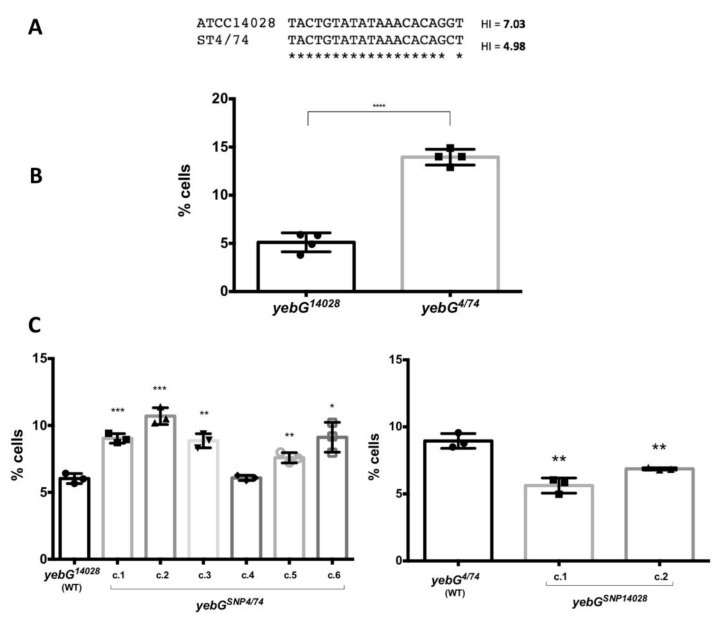
Comparison of YebG^ON^ subpopulation sizes between *S.* Typhimurium strains ATCC 14028 and ST4/74. (**A**). Single-nucleotide polymorphism in the *yebG* SOS boxes of ATCC 14028 and ST4/74. HI values are indicated. (**B**). Percentages of YebG^ON^ cells in ATCC 14028 and 4/74. (**C**). Percentages of YebG^ON^ cells in in ATCC 14028 and 4/74 derivatives whose SOS boxes have been swapped. Statistically significant differences are indicated (*p*-value; * *p* < 0.05, ** *p* < 0.01, *** *p* < 0.001, **** *p* < 0.0001).

**Figure 7 cells-10-00943-f007:**
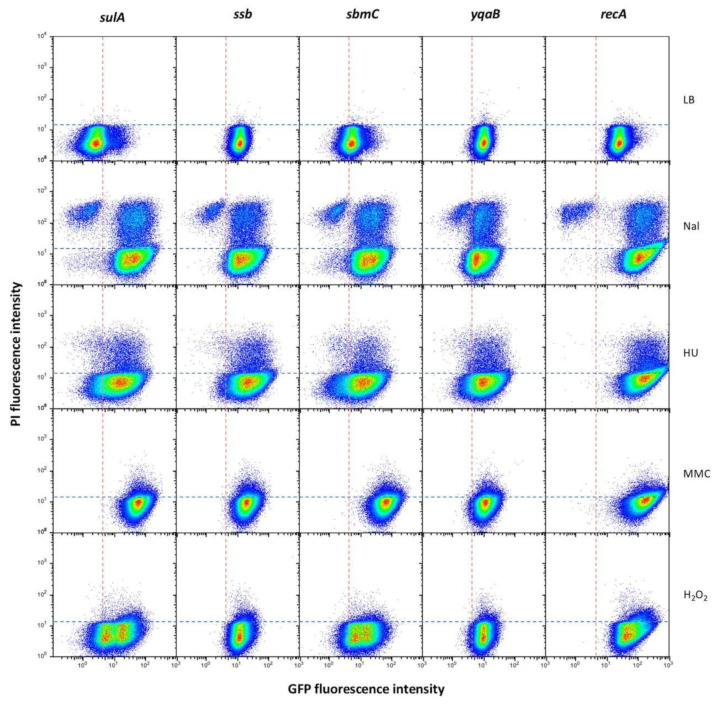
Expression patterns of genes belonging to groups I (*sulA*), II (*ssb*), III (*sbmC*) and miscellaneous (*yqaB* and *recA)* upon SOS induction. The DNA damaging agents used were nalidixic acid (Nal), hydroxyurea (HU), mitomycin C (MMC) and hydrogen peroxide. The vertical division sorts OFF (GFP^−^, left) and ON (GFP^+^, right) subpopulations. The horizontal division sorts live (PI^−^, bottom) and dead (PI^+^, top) subpopulations.

**Figure 8 cells-10-00943-f008:**
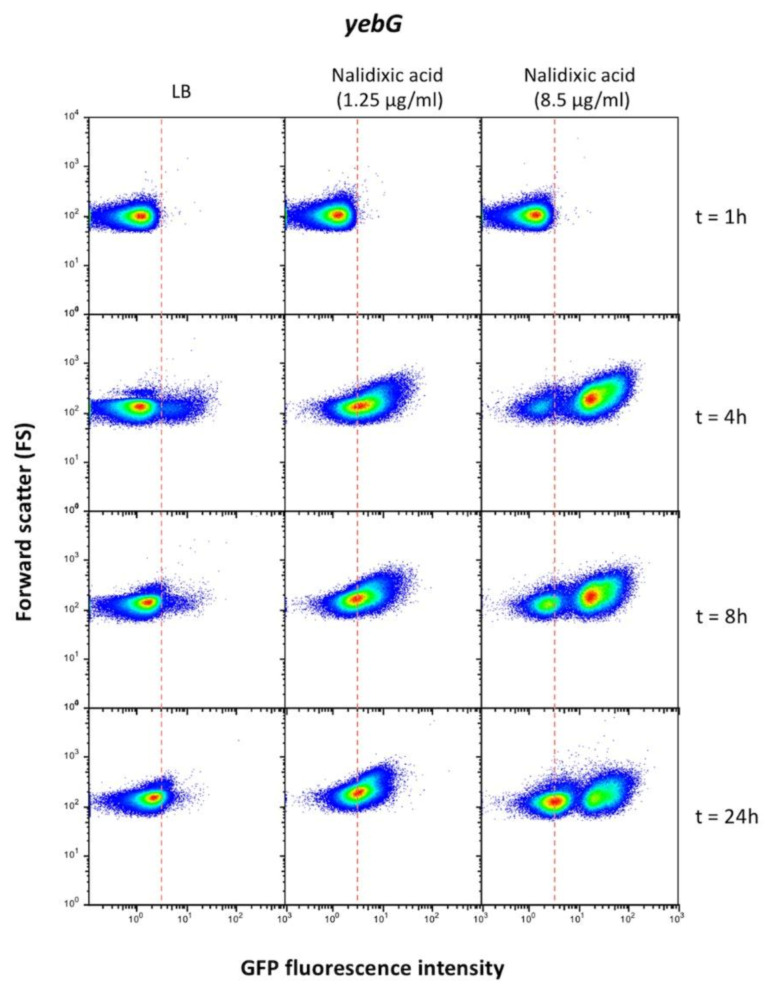
Single cell analysis of *yebG* expression in the presence of low and high concentrations of nalidixic acid. The time since nalidixic acid was added to the culture is indicated. The dashed line shows the boundary between OFF and ON states, established as the expression level of a nonfluorescent sample (Appendix A).

**Figure 9 cells-10-00943-f009:**
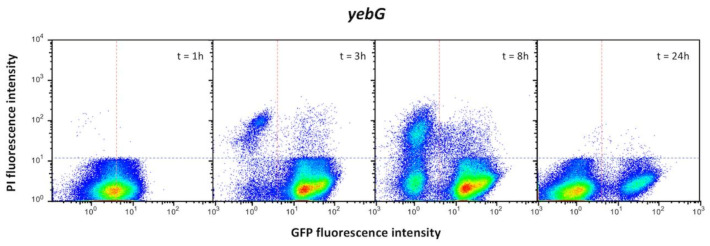
Single cell analysis of live and dead YebG^OFF^ and YebG^ON^ subpopulations after addition of nalidixic acid. The vertical division sorts OFF (GFP^−^, left) and ON (GFP^+^, right) subpopulations. The horizontal division sorts live (PI^−^, bottom) and dead (PI^+^, top) subpopulations.

**Table 1 cells-10-00943-t001:** SOS response genes of *S. enterica* identified as loci that harbor putative LexA boxes and are upregulated in the presence of nalidixic acid.

Gene ID	Gene Name	Fold Change	Number of SOS Boxes	Distance to ATG (bp)	Heterology Index *
STM14_4584	*tisB*	610.09	1	236	6.13
STM14_2287	*yebG*	41.34	1	17	7.03
STM14_3289	*recN*	39.41	3	8, 26, 48	12.97, 7.72, 8.82
STM14_3417	*recA*	35.71	1	64	5.92
STM14_1215	*sulA*	26.70	1	21	1.49
STM14_3210	*dinI* ^Gifsy-1^	23.20	1	19	4.55
STM14_1331	*dinI*	22.82	2	19, 42	4.00, 17.60
STM14_2423	*umuD ***	20.10	1	15	4.38
STM14_4775	*yigN*	19.46	1	44	8.71
STM14_3002	*cysP*	15.47	1	65	18.16
STM14_4846	*nlhH*	15.11	1	13	7.63
STM14_1156	*dinI* ^Gifsy-2^	14.46	1	19	3.92
STM14_0926	*uvrB*	14.13	1	73	5.66
STM14_4752	*uvrD*	13.87	1	105	11.48
STM14_0369	*dinP ****	11.30	1	15	6.80
STM14_5112	*uvrA*	11.16	1	80	6.99
STM14_953	*dinG*	10.61	1	14	10.08
IstR-1,2	*istR-1,2*	10.26	1	40	6.13
STM14_5114	*ssb*	9.63	1	24	6.99
STM14_4847	*higB-2*	8.05	1	184	7.63
STM14_4236	*dinJ*	7.82	1	14	8.27
STM14_3214	*--*	7.70	1	6	12.33
STM14_5094	*lexA*	6.86	2	6, 27	14.48, 7.94
STM14_3568	*gudD*	5.43	1	0	16.97
STM14_3405	*yqaB*	5.32	1	12	16.26
STM14_1439	*dinI^Gifsy-3^*	4.56	1	19	5.26
STM14_2752	*yejK*	4.54	1	102	20.89
STM14_2422	*umuC* **	3.91	1	--	4.38
STM14_2753	*yejL*	3.74	1	57	20.89
STM14_2648	*thiM*	3.62	1	177	18.02
STM14_3627	*mutH*	3.58	1	228	12.99
STM14_2551	*sbmC*	3.55	1	21	6.32
STM14_1492	*msgA*	3.44	1	14	11.88
STM14_2650	*rcnB*	3.38	1	212	18.02
STM14_1589	*ydjQ*	3.33	1	0	6.69
STM14_0161	*yacA*	3.30	1	403	19.46
STM14_4344	*yhjE*	3.05	1	202	14.01
STM14_5011	*hupA*	2.74	1	190	15.31
STM14_1432	*--*	2.71	1	6	12.33
STM14_4158	*slyX*	2.70	1	146	21.28
STM14_3283	*corE*	2.54	1	98	20.11
STM14_1385	*ndh*	2.37	1	157	12.04
STM14_0374	*frsA*	2.33	1	360	13.12
STM14_5490	*deoD*	2.21	1	210	18.78
STM14_5518	*--*	2.19	1	71	17.26
STM14_1605	*ydjM*	2.11	2	14, 32	9.12, 12.85

* Defined in Section 3.4. ** *umuD* and *umuC* form and operon, and their SOS box is located upstream of *umuD.* *** Homologue of *E. coli dinB.*

## Data Availability

The data that support the findings of this study are available from the corresponding author upon request.

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
