# Peer review of "Genome-Wide Identification and Expression Analysis of SOS Response Genes in *Salmonella enterica* Serovar Typhimurium"

_cells, 2021, doi:10.3390/cells10040943_

Round 1

Reviewer 1 Report

The manuscript by Ángela Mérida-Floriano and Josep Casadesús provides new insight into the SOS regulon in Salmonella. The study is solid and of significant interest considering the wide conservation and importance of the SOS response throughout the bacterial kingdom. I have just a few points for the authors to address:

  1. Identification of SOS regulated genes by transcriptomics would have benefited from a control using the LexA Ind- mutant strain. This would ensure that genes which become upregulated after DNA damage in an SOS-independent manner would not be included in the candidate list, and could help to relax the stringent criteria of the presence and distance of a LexA box that the authors applied to narrow down the candidate list. Without such stringent criteria, it may be possible to identify additional SOS regulated genes.
  2. Can the authors comment on the large number of downregulated genes, which is almost as large as the number of upregulated genes?
  3. Flow cytometry figures 1,2,3 show dashed lines, but it is not clear what these lines refer to and how they were chosen. Please clarify.
  4. Section 3.3: I found the distinction of genes into groups 1,2,3 interesting but I wonder if this distinction can be made more rigorous and quantitative, e.g. based on the abundances of the ON and OFF populations and the shift in expression between the states? Do the groups actually fall into separate clusters if analysed in that way?
  5. Line 311: “A tentative interpretation is either that SOS induction did not take place in such cells or that SOS induction failed to tolerate DNA damage.” Since this population is GFP- (no SOS response) then the interpretation that SOS induction failed to tolerate DNA damage cannot be true?
  6. Paragraph staring in line 337: Can the absence of dead cells after 24h treatment be explained by cell lysis or excessive filamentation such that these cells would not be detected in flow cytometry? Regarding the hypothesis that live cells that are SOS negative might be resistant mutants, can the authors plate cells after 24 treatments and test if they have obtained genetic resistance?
  7. Lines 388 ff: Discussion about the unexpected relation between LexA binding affinity and SOS expression should also feature results from this paper, which linked LexA affinity and induction timing of different SOS genes: Culyba et al. Non-equilibrium repressor binding kinetics link DNA damage dose to transcriptional timing within the SOS gene network. PLOS Genetics 14, e1007405 (2018).
  8. Lines 425 ff: The point about analogue regulation of the SOS response is interesting, and is supported by experiments that show a continuous scale of LexA degradation. See Jones et al. Imaging LexA degradation in cells explains regulatory mechanisms and heterogeneity of the SOS response. bioRxiv 2020.07.07.191791 (2020)

Reviewer 2 Report

This paper by Merida-Floriano et al titled “Genome-wide identification and expression analysis of SOS response genes in Salmonella enterica serovar Typhimurium” describes a screen for SOS response genes based on bioinformatic and flow cytometry methods. Initially, candidate SOS response genes were identified if a 20 bp LexA binding sequence is located near the gene. This approach identified over 300 candidate genes, some of which had been previously characterized as members of the SOS regulon. RNAseq experiment was then carried out to narrow down this list of SOS response candidatesFinally, candidates were GFP tagged and flow cytometry was used to track gene expression in S. enerica populations in normal and DNA damage inducing conditions.  

The flow cytometry experiments (by the authors own admission) generated results that were confusing and somewhat inconsistent with expectationsThe authors do a commendable job in providing well considered interpretations of the complex and unexpected observations provided by their experiments. However, in my opinion the observation that many of the candidate SOS regulon genes are activated independently of experimentally induced DNA damage, compromises the RNAseq data that was used to identify SOS response genes in the first placeThat many of the cells in the control group (for the RNAseq) are presumably displaying an active SOS responsewould raise questions about the validity of using upregulation in the treatment group to define SOS response genes. If this is true then many of the conclusions based on subsequent experiments are called into question. 

Given that the results of the flow cytometry are somewhat unexpected, I felt that the authors needed to do more to directly address the possibility that the observations were due to experimental artifactsAdding to my uncertainty, many of the figure legends are incomplete, missing key details that would allow me to properly evaluate the authors conclusions. On line 433 of the Discussion the authors state "an incontestable conclusion may be that phenotypic heterogeneity adds an additional layer of complexity to the SOS response" - I couldn’t agree more. For this reason, I do hope that the authors are able to publish this work at some stage and I hope that my critical feedback is helpful for this purpose. 

Line 70-74: I am not familiar with Salmonella culturing methods, but it is not clear from this description if the ‘non-inducing’ and SOS-inducing' cells were both in stationary phaseThis should be clarified as any differing growth conditions (apart from the addition of nalidixic acidbetween the two groups would add an additional layer of complexity to the interpretation of the RNAseq data. 

Line 77: “cDNA from the different samples was pooled”. This statemenis confusing because it implies that cDNA from the biological replicates was pooled together before library preparation and sequencing. I assume the authors mean to say that the indexed sequencing libraries were pooled; this approach would allow the biological replicates to be recovered via demultiplexing after sequencing. If my assumption is correct the reference to ‘pooling cDNA’ should be removed to avoid any confusion here.  

Line 88: “The complete RNA-seq pipeline used for this study is described in https://github.com/will-Rowe/rnaseq". Technically this is not the “complete pipeline” because it does not include the “Degust” analysis workflow. The code in this repository seems to relate specifically to the generation of per-gene read counts. Also, as far as I can tell the author of this RNAseq repository (Dr Will Rowe) is not associated with this study, thus Dr Rowe is under no obligation to maintain this repository in its current stateFor the sake of reproducibility, I would strongly recommend that the authors clone this repository into their own GitHub account and tag the pipeline with a version number that can be referenced in the manuscript. 

Line 99: “log2FC <=1. This permissive cut-off..". I assume this is a typo, should it be log2FC >=1? A log2FC cuff-off of less than 1 would discard all the genes that are strongly differentially expressed in response to the treatment? On line 143 this value suddenly changes to “log fold change >2”. So which cut-off was used? I don’t think you need to describe the cut-off value as permissive given these values are somewhat arbitrary.  

Lines 136-142: Please add information on the per-sample sequencing yield obtained (average number of reads per sample) as this information represents the sample size or sensitivity of the RNAseq experiment. 

Line 155 “Genes that were not known to be part of the SOS regulon were also identified as loci under LexA control”. I would recommend adding “putative” or “candidate” to reflect that this experiment does not provide any direct evidence that these genes are transcriptionally controlled by LexA (it could be a secondary response). RNAseq data only shows a correlation between expression and a treatment condition, it does not provide direct evidence of specific regulatory control. 

Table 1. I assume the “ID” column is the “Gene ID” or is it some other “ID” (this should be clarified)? Is “fold Change” raw fold change or log fold change (this should be clarified)? At this point of the paper the “heterology Index” has not been described, it would be good to point out in the table description that this is introduced in section 3.4.  

Section 3.3. That these putative SOS-response genes appear to be activated independent of “known DNA damaging agent” raises the question that these observations could be experimental artifacts. Admittedly, flow cytometry is not my area of expertise, but the authors do noappear to address this possibility directly by presenting data from experimental controls (or other evidence) that would argue against this conclusion.  

Figure 1. The legend lacks the required information for me to critically interpret this figure. What does the red dashed line indicate? What do the different colors represent? The figures should be interpretable without needing to refer to the main text. Detail on the separation lines are finally provided in Figure 7, however the methodology/criteria used to define the position of this line appears to be missing (at least from section 2.8). As the placement of this line defines the Three main gene groups” (line 177) it is important that this is properly describedOnly rows 2 and 3 are labelled. The percentage of cells in the ON population is shown in the plots on the top row, but this is not described in the legend. Most of these comments apply to the other flow cytometry figures as well. 

Figure 1. Flow cytometry is not my area of expertise, but for several genes (ie umuC, STM14_3214, uvrAin the lexA and recA deletion backgrounds, GFP signal can clearly be observed on the right side of the red dashed line. What is the explanation for this observation given the authors declare that In agreement with this view, our analysis of gfp fusions did not detect ON cells in a [delta] recA background” (line 187)? 

Figure 3. Again, flow cytometry is not my area of expertise, but Isn’t the LexA expression in the LexA deletion background a negative control? If so why was this plot left out of the figure (column 4, row 2)? 

Line 191-193 “Furthermore, the size of the ON subpopulation increased in a recD background and to a lesser extent in a recF background (Fig. S2)”. No mention is made of the fact that this conclusion is based on data from only one gene (sulA). 

Figure 4. I assume the asterisk(*) represent the results of a statistical test? If so what test was used? What is the difference in the test result with one versus two asterisk, this information is absent in the legendThis latter information is eventually appears in figure 6, but it should be defined when first used. There are 10 data points plotted in the LexA-dependent plots in Figure 4A and B. But there is data for 12 genes in the LexA-dependent bi-stability group in Figure 1; why are some of these genes excluded in Figure 4? Could the authors clarify this? 

Figure 5. As above, there are 11 genes plotted as part of the correlation for the “ON subpopulations of genes with LexA-dependent bistability”, yet the same group consists of 12 genes in Figure 1. I think this difference requires some explanation. 

Figure 6. What statistical test was used? 

Figure 8. The dashed ON/OFF delineation lines are no longer included in this figure as they were for the other flow cytometry plots. There needs to be consistency in the way the data is presentedor at least some explanation as to why this was changed. 

Minor points 

“ATCC 14028” and “ATCC 14,028 is used interchangeable throughout the manuscript, can the authors be consistent here. It is unusual for strain designations to include commas, so I would favor the use of the former version throughout the manuscript. 

The interchangeable use of “recA background” (line 184) and "delta recA background” (line 188) is confusing, the latter is much more descriptive. 

Reviewer 3 Report

The manuscript describes an interesting and useful set of experimental data on the genome-wide analysis of SOS-response in Salmonella enterica. The results of this study were obtained by carefully planned and executed work. However, there are a number of minor issues which should be addressed in the revised manuscript.

Table 1. It does not follow from the text what the meaning of the heterology index. It should be explained in the text. Besides, I was confused finding in this table several values of this index. What does it mean? The range or 95% confidence intervals? It should also be explained.

Reviewer 4 Report

In this study, authors combine bioinformatic and transcriptomic analysis to identify genes belonging to the S. enterica SOS regulon. They further study genes listed as canonical, i.e. harboring LexA boxes and activated by SOS inducing conditions.  While the study is descriptive, it highlights bistability and heterogeneity of gene expression for genes related to a stress response, in the absence and presence of stress. Such heterogeneity is important to be addressed and decribed in details as it was done here, and will be of use to future studies of stress related gene expression.

Interestingly bistability (fluorescent subpopulation) can be observed in the absence of stress, this is porposed to be triggered by activation of the SOS response upon spontaneous DNA strand breakage, which is an interesting point.

Importantly, SOSOFF and SOSON subpopulations, were observed both in the absence of DNA damage and upon SOS response activation, showing that whole population studies showing mean expression of genes in a population (e.g beta gal tests or RNA-seq), need to be combined with approaches allowing the detection of subpopulations.

The manuscript is clearly written and presented.

I have minor comments and questions:

  • Could the authors comment on the interpretation of flow cytometry experiments in relation to filamentation, particularly upon SOS induction by inducers such as fluoroquinolones or MMC ? did they use a non-filamenting genetic context (such as how it is done in an E. coli delte-sfiA mutant)? Were the concentrations of inducers causing major filamentation? The flow cytometry data shows fluorescence of GFP against PI for SOS inducing conditions, but not SSC or FSC, filamentation (or its absence) is thus not shown here.
  • One interesting point is also that genes that were not known to be part of the SOS regulon were also identified as loci under putative LexA control, for example cysP. Not directly relevant for the point of this paper focusing on genes with an SOS box, but are there any indirectly regulated genes (non-canonical, not harboring sos box) strongly regulated by SOS, such as those that were described in PMID: 29783948?
  • The SOS gene ruvA was manually added to the list as it had not appeared among the loci upregulated by nalidixic acid”, could this be due to the timing of induction? Transient induction?
  • Same question for the following statement: “After preliminary validation, certain genes were excluded from further analysis due to lack of fluorescence (dinG, ydjQ and ydjM) or because flow cytometry failed to detect induction by nalidixic acid (cysP and gudD).”

Also, how stable was the GFP used for this study: would the fluorescent protein be degraded in the case of transient expression, or would the fluorescence accumulate with time?

  • The question of GFP stability is also relevant for the interpretation of GFP off PI off cells, i. e. viable cells which do not induce SOS, is it because they have never induced SOS or because SOS was transiently induced not detected after damage repair, stable GFP would still be detected upon transient induction but not unstable GFP.

“One is that repair of DNA damage is highly efficient in such cells, thus permitting that the SOS response is rapidly turned off.” Such a rapidly turned off expression could be observed only if GFP is rapidly degraded. 

One way to experimentally address this would be to use a control with constitutive GFP expression. I any case, I believe information regarding the stability of the GFP used in this study should at least be mentioned in the manuscript. 

Author Response

File attached

Round 2

Author Response

File attached
